# Two-step light gradient boosted model to identify human west nile virus infection risk factor in Chicago

Guangya Wan[1,2], Joshua Allen[1], Weihao Ge [1]*, Shubham Rawlani[1,3], John Uelmen [4], Liudmila Sergeevna Mainzer [1,5], Rebecca Lee Smith [1,4,5]*

1 National Center for Supercomputing Applications, University of Illinois, Urbana-Champaign, Illinois, United States of America, 2 Department of Statistics, University of Illinois, Urbana-Champaign, Illinois, United States of America, 3 Information School, University of Illinois, Urbana-Champaign, Illinois, United States of America, 4 Department of Pathobiology, University of Illinois, Urbana-Champaign, Illinois, United States of America, 5 Car R. Woese Institute for Genomic Biology, University of Illinois, Urbana-Champaign, Illinois, United States of America

* wge2@illinois.edu (WG); rlsdvm@illinois.edu (RLS)

**Data Availability Statement:** Data cannot be shared publicly because of privacy concerns. Data are available from the Illinois Department of Public

## Abstract

West Nile virus (WNV), a flavivirus transmitted by mosquito bites, causes primarily mild symptoms but can also be fatal. Therefore, predicting and controlling the spread of West Nile virus is essential for public health in endemic areas. We hypothesized that socioeconomic factors may influence human risk from WNV. We analyzed a list of weather, land use, mosquito surveillance, and socioeconomic variables for predicting WNV cases in 1-km hexagonal grids across the Chicago metropolitan area. We used a two-stage lightGBM approach to perform the analysis and found that hexagons with incomes above and below the median are influenced by the same top characteristics. We found that weather factors and mosquito infection rates were the strongest common factors. Land use and socioeconomic variables had relatively small contributions in predicting WNV cases. The Light GBM handles unbalanced data sets well and provides meaningful predictions of the risk of epidemic disease outbreaks.

## Introduction

West Nile Virus (WNV) is a mosquito-borne flavivirus that has been circulating in the United States for two decades, first appearing in New York in 1999 [1–3]. The disease is spread in an enzootic mosquito-bird-mosquito circulation [4–7], and zoonotic transmission occurs when humans are bitten by a WNV-positive mosquito [8]. Because there are no vaccines for WNV in humans, prediction of WNV-positive mosquitoes is used to inform public health actions to clear mosquitoes in areas of high risk [9] and to warn the general public of increased risk.

Efforts have been made to build predictive models of WNV spread and identifying the predictive factors [10]. Predicting human cases would help to identify high-risk populations, and therefore enable protective measures. The foremost predictors are the temperatures and precipitations. Paz [11] analyzed major weather factors and found temperature and precipitation

Health Institutional Data Access / Ethics Committee, which has imposed these restrictions for researchers who meet the criteria for access to confidential data. Contact Megan Patel, Megan. Patel@illinois.gov (Surveillance and Informatics Epidemiologist in the Office of Health Protection, Division of Infectious Disease), for information about data access.

**Funding:** The first author was supported by the Students Pusing Innovation Internship (SPIN) at the National Center for Supercomputing Applications. https://spin.ncsa.illinois.edu/ The project was funded by the NCSA Center-Directed Discretionary Research (CDDR).

**Competing interests:** The authors have declared that no competing interests exist.

are associated with WNV human cases. A temperature range of 10–35˚C is advantageous for mosquito breeding activity. However, an association of temperature with WNV infection risk is not always positive. Hahn et.al.[12] performed a climate-region-wise analysis. They have found that on the national scale and in most regions (except for southwest, west, and northwest climate regions), above-average temperature increases WNV risk. Shocket et al. [13] has identified the optimal temperature range for mosquitoes that vector WNV is between 23–26˚C. Precipitation and humidity have complex associations with mosquito population and infection rate, as well. Interaction between temperature and precipitation also explains a significant part of the WNV mosquito infection rate [14]. Poh et al. identified that temperature and rainfall increase mosquito abundance [15].

In addition to temperature and precipitation, other factors such as humidity and wind velocity affect mosquito abundance [16]. Peper et al. have studied WNV and mosquito surveillance records from Lubbock, TX, and have found that the probability of mosquito infection depends on the weather variables including the time in the year, wind, visibility, humidity, dew point, and the time lag of these variables [17]. They also found that weather has a temporal autocorrelation, which brings lagging effects into play [18, 19]. DeFelice has discussed the lag in reporting of both mosquito infection and human cases that reduces real-time WNV forecast accuracy and proposed recursive optimization and Poisson process simulation for the retrospective forecast to solve the problem [20].

The landscape also contributes to WNV risk. Studies have identified land cover factors such as vegetation, urbanization, mosquito breeding sites, and wetlands to be associated with WNV incidences [21–23]. Sánchez-Gómez et al. have discussed how temperature and the presence of wetlands influence WNV circulation in vectors and humans [21]. Hernandez et al. have identified weather, demographic, and controlling measurements including temperature, precipitation, ethnicity, mosquito breeding sites, targeted prevention, and education as key predictors, where the mosquito breeding sites are associated with land cover [22] Myer and Johnston have analyzed a 15-year span of data in Nassau County, NY, and identified landscape factors including high normalized difference vegetation index (NDVI), wetlands, and high urban development have a negative association with WNV incidences [23]. Farooq et.al. have estimated WNV expansion risk and found early spring weather, population, and agriculture activities can be important factors for early warning systems to predict Europe WNV outbreak [24].

Demograpic disparities are also observed in the previous studies. In [22], Hernandez found that in addition to weather and landscape, ethnicity, targeted prevention, and education are key predictors. Especially, Hernandez et al. have pointed out that increased percentage of White people in the census tract is associated with the incidences of WNV cases might be related to underreporting in other ethics group due to differences in health insurance and willingness to seek medical care, resulting in under-reporting of other ethnic groups. Additionally, the ethnical difference in WNV risk could be associated with a behavioral risk factor, such as whether individuals work outside the home, which might increase the chance of contracting WNV. Bassal et.al. investigated demographic disparities for WNV IgG levels in Israel and identified different WNV seroprevalence among geographical regions. Bassal et. al. also discovered different prevalence among racial groups, which have different socioeconomic status [25].

Linear regression and ensemble tree methods are the two most commonly used approaches for predicting WNV incidence or mosquito populations. Hernandez et al. started with chi-squared tests to identify a list of candidate factors and then used regression to find the strongest predictors [22]. Karki et al. used a stepwise model selection procedure to automatically test all factors and find the strongest predictors [26]. However, the risk of WNV is not linear with the factors. Furthermore, linear models have high specificity and perform best when

there are no cases of viral infection, but have poor sensitivity when there are cases (low recall). To address these two issues, we instead of use the light gradient boosting method (GBM) [27] to build trees that selects the features splitting the categories best. Previously, the ensemble methods, especially the random forest approach is also widely used [add citations, 28, 29]. LightGBM and random forest are different in the following ways. LightGBM is a gradient boosting decision tree algorithm, while random forest is an ensemble learninbg method based on decision trees. Therefore, lightGBM trains decision trees in a sequential way, where the learning rates are derived from the errors from the previous trees. On the other hand, random forest learns with averaging or voting. Moreover, lightGBM uses a greedy algorithm that grows trees with a leaf-wise strategy, while random forest creates a more balanced tree with a depth-wise strategy. As a result, lightGBM has higher efficiency, especially when the data set is large or feature space dimension is high. We performed a two-step light GBM approach as recommended for other ensemble tree methods [28, 29]. In the first step, all factors are included in the model. And then a second light GBM classification/regression is performed based on the top factors selected by the first model [28].

We have hypothesized that, in addition to natural factors such as mosquito infection rate (MIR), weekly temperature, temperature in January, and precipitation, socioeconomics and land cover factors will also be predictive factors for the WNV occurrences. We also hypothesized that natural factors might have lagging effects. These effects, linear or not, can be detected by the light GBM approach and identify areas at high risk of WNV cases and provide guidance for health intervention.

## Methods

### Data set and pre-analysis

The dataset we used is described in more detail in Karki, et al. [26]. The dataset includes the number of human disease cases from 2005–2016 in Cook and DuPage Counties, IL, as the dependent variable, and several independent variables comprising weather, socioeconomic, land cover, and mosquito infection rates (MIR). All variables were aggregated on a weekly temporal resolution and on a spatial grid of 1 km wide hexagons for the study region.

The human disease data is described as a binary number that represents whether a case occurs in a hexagon in a given week. We performed the two-sample Kolmogorov-Smirnov (KS) test [30] and the two-step light GBM classification [27] to build the model to predict the human illness data and to derive the illness probability from the model.

Weather variables include temperature and precipitation, as well as the lagged variables representing temperature and precipitation 1 week, 2 weeks, 3 weeks, and 4 weeks before human case report date. The original weather data was collected by PRISM [31], on 4km grid. The weather data are then mapped to hexagons by Karki, et al. [26] The land cover data include urban areas (developed open space, developed low intensity, developed medium intensity, developed high intensity), forest (deciduous, evergreen, and mixed), barren land, shrubs, grassland, pasture, cultivated crops, woody wetlands, herbaceous wetlands, and open water. Karki, et al.[26] retrieved the land cover data from the 2016 National Land Cover Database (NLCD) [32] and aggregated the percentage of different land covers in the hexagons.

For the socioeconomic data used by Karki, et al. [26], the 2016 census data from the US Census Bureau [33] was applied across all years. The data were converted from the census tract level to the hexagon level by assuming homogeneous socioeconomic status within each census tract. To determine the sensitivity of the socioeconomic data to annual changes, we replicated the mapping procedure with the 5-year rolling averages from 2010–2017 and performed the model analysis with both datasets (S1, S2 Files). We found that the results are similar, and the

conclusions do not change; therefore, we will present the model built with the 2016 census data.

The variables we used are listed in Table 1 below.

We applied the Kolmogorov-Smirnov (KS) test [30] to assess differences in feature distribution between WNV_binary = 1 groups (the groups have WNV cases) and WNV_binary = 0 groups (the groups have no WNV cases). The KS test is distribution-free. It is advantageous because it doesn't rely on distribution assumptions and can reveal important discriminating features in various data types.

Features with low $p$-values from the KS test indicate significant differences between the WNV_binary = 1 and 0 groups, making them important for explaining variations in the dependent variable. $P$-values represent the likelihood of observed distribution differences occurring by chance. We calculated the -log($p$-value) so that the larger the -log($p$-value), the more significant feature importance will be. Features with larger -log($p$-value) are considered as important for predicting the WNV_binary values.

The KS test itself does not deal with any correlations between the features. We then assess the collinearity by generating the covariance plots calculated from Pearson's correlation between the features. The features with correlation values > = 0.35 are considered correlated. Among the correlated features, we keep the ones with the largest -log($p$-values) from the KS test, i.e., the most important ones predicted in the distribution-free test. Therefore, the features we kept are independent from each other, ready for modeling.

## Two-step light GBM modeling

The hyperparameter for the light GBM is tuned with randomized search with a predefined set, evaluated on the metric log-loss score as the decision criterion, which can help deal with the highly zero-inflated characteristic of the WNV case number. After randomized search, the code automatically applied the best hyperparameter set to run the model. We used the lightgbm package in Python [34] to perform the light GBM method. Table 2 shows the hyper-parameter distribution we used.

The model was built using a heuristic approach with two light GBM categorization procedures. After removing the correlation, we ran the first light GBM procedure on all remaining variables. We then examined the distribution of feature importance, selected the top variables by the natural gap in the distribution, and ran another light GBM procedure. Feature importance is defined as the mean decrease in impurity when a given feature is included to split the WNV_binary = 0 and WNV_binary = 1 cases. Feature importance is represented by the negative logarithm of the absolute value of importance. We evaluated the receiver operating characteristics area under the curve (ROC-AUC) to find the best threshold for a minimum model. The ROC-AUC score is insensitive to imbalanced data. With the threshold identified, we are able to evaluate the accuracy, recall, precision, and F-1 score [35]. We first fit the model with high and low income data to confirm that the models are similar (S3 File). Therefore, we build our model based on the full dataset. We then examine the distribution of feature importance and select subsets of features to build reduced models. We examine the performance of the reduced models to find a minimal model that retains predictive power.

Then, in the final model, we evaluated the relative importance of the covariates to identify important predictive features for WNV cases in our models. For the features of interest, we generate partial dependence (PD) plots to show their marginal predicted probability. The slope of the PD plot represents the strength of the feature. The shape of the PD plot could also indicate whether the effect is monotonic. The PD plots could easily show the nonlinear effects that are difficult to identify by regression.

**Table 1. List of variables involved in building the models.** We have variables representing Land cover, Mosquito infection rate, Weather, and Demographics factors.

| Variables | Notation |
|---|---|
| *Land cover* | |
| Proportion of developed open space | dospct |
| Proportion of developed low intensity | dlipct |
| Proportion of developed medium intensity | dmipct |
| Proportion of developed high intensity | dhipct |
| Proportion of deciduous forests | dfpct |
| Proportion of evergreen forests | efpct |
| Proportion of mixed forests | mfpct |
| Proportion of barren land | blpct |
| Proportion of shrubs | shrubspct |
| Proportion of grassland | glandpct |
| Proportion of pasture | pasturepct |
| Proportion of cultivated land | clpct |
| Proportion of woody wetlands | wwpct |
| Proportion of herbaceous wetlands | hwpct |
| Proportion of open water | owpct |
| *Mosquito infection rate* | |
| Weekly average mosquito infection rate | mir_mean |
| Mosquito infection of one week before | mir_lag1 |
| Mosquito infection of two weeks before | mir_lag2 |
| Mosquito infection of three weeks before | mir_lag3 |
| Mosquito infection of four weeks before | mir_lag4 |
| *Weather* | |
| *Temperature* | |
| Weekly average temperature | temp_c |
| Average temperature one week before | temp_lag1 |
| Average temperature two weeks before | temp_lag2 |
| Average temperature three weeks before | temp_lag3 |
| Average temperature four weeks before | temp_lag4 |
| Temperature in January | tempJan |
| *Precipitation* | |
| Weekly average precipitation | preci |
| Average precipitation one week before | preci_lag1 |
| Average precipitation two weeks before | preci_lag2 |
| Average precipitation three weeks before | preci_lag3 |
| Average precipitation four weeks before | preci_lag4 |
| *Demographic factors* | |
| Percentage of White population | whitepct |
| Percentage of African American population | blackpct |
| Percentage of Asian population | asianpct |
| Population of Hispanic | hispanicpct |
| Total population | tot_pop |
| Median household income | income |
| Percentage of houses built before WWII | hpctpreww |
| Percentage of houses built after WWI | hpctpostww |
| Percentage of houses built between 1970–1989 | hpct7089 |
| Percentage of houses built after 1990 | hpctpost90 |

**Table 2. List of hyperparameters to optimize in the lightGBM models.**

| parameters | meaning | values |
|---|---|---|
| **num_leaves** | number of leaves | randint (12,20) |
| **min_child_samples** | minimal data in each leaf | randint (40,100) |
| **min_child_weight** | the sum of hessians (second order derivative of the loss function w.r.t. the current prediction value) of the data points in the leaf [cite: https://github.com/microsoft/LightGBM/issues/3816] | [1e-2, 1, 5] |
| **subsample** | bagging fraction | U(0.75, 1) |
| **colsample_bytree** | subsampling columns when contructing each tree | U(0.8, 0.95) |
| **reg_alpha** | l1 regularization | [0, 1, 5, 10] |
| **reg_lambda** | l2 regularization | [0, 1, 5, 10] |

# Results

## KS test

We performed univariable KS tests on all variables (Fig 1). We found that temperatures and mosquito infection rates were significantly related to the WNV risk in the model. On the other hand, precipitation, land cover and socio-economic characteristics were not significantly related to WNV risk. The variables are listed in S4 File.

## Variable correlations

Fig 2 shows the correlation between the variables. We found that weekly temperatures have a strong positive temporal correlation (0.47–0.84). On the other hand, the lagged effects of weekly MIR (0.075–0.18) and weekly precipitation (-0.022–0.044) are not as strongly correlated. Weekly MIR and weekly precipitation are also independent of other variables.

We also found that income is strongly correlated with race. Income has a high positive correlation (0.54) with the white race percentage in the hexagon area, and a medium-high negative correlation with the black race percentage (-0.46) and the Hispanic race percentage (-0.37). The white and black population percentages have a strong negative correlation with each other (-0.87), which is to be expected since the total population percentages should add up to 100%.

For each set of medium to highly correlated variables, we kept the variables with the highest KS scores for the light GBM analysis. The remaining variables are: All precipitation and MIR variables, mean temperature of 4 weeks before the human case report, mean temperature in January, total population, proportion of developed low intensity, proportion of open water, proportion of barren land, proportion of evergreen forest, proportion of shrubs, proportion of grassland, proportion of pasture, proportion of cultivated land, proportion of woody wetlands, emergent herbaceous wetlands, percent temperature in January, house post World War II, and income.

## Light GBM based on all selected features

We built our models using cross-validation, randomly splitting training and test sets, and then selected the best parameter based on the log-loss criteria. The Gini feature importance of each predictor in the model is shown in Fig 3, and its performance on the test set is shown in Table 2. The best model selected during the process has min_child_samples = 72, min_child_weight = 5, num_leaves = 14, reg_alpha = 10, reg_lambda = 10, and subsample = 0.763.

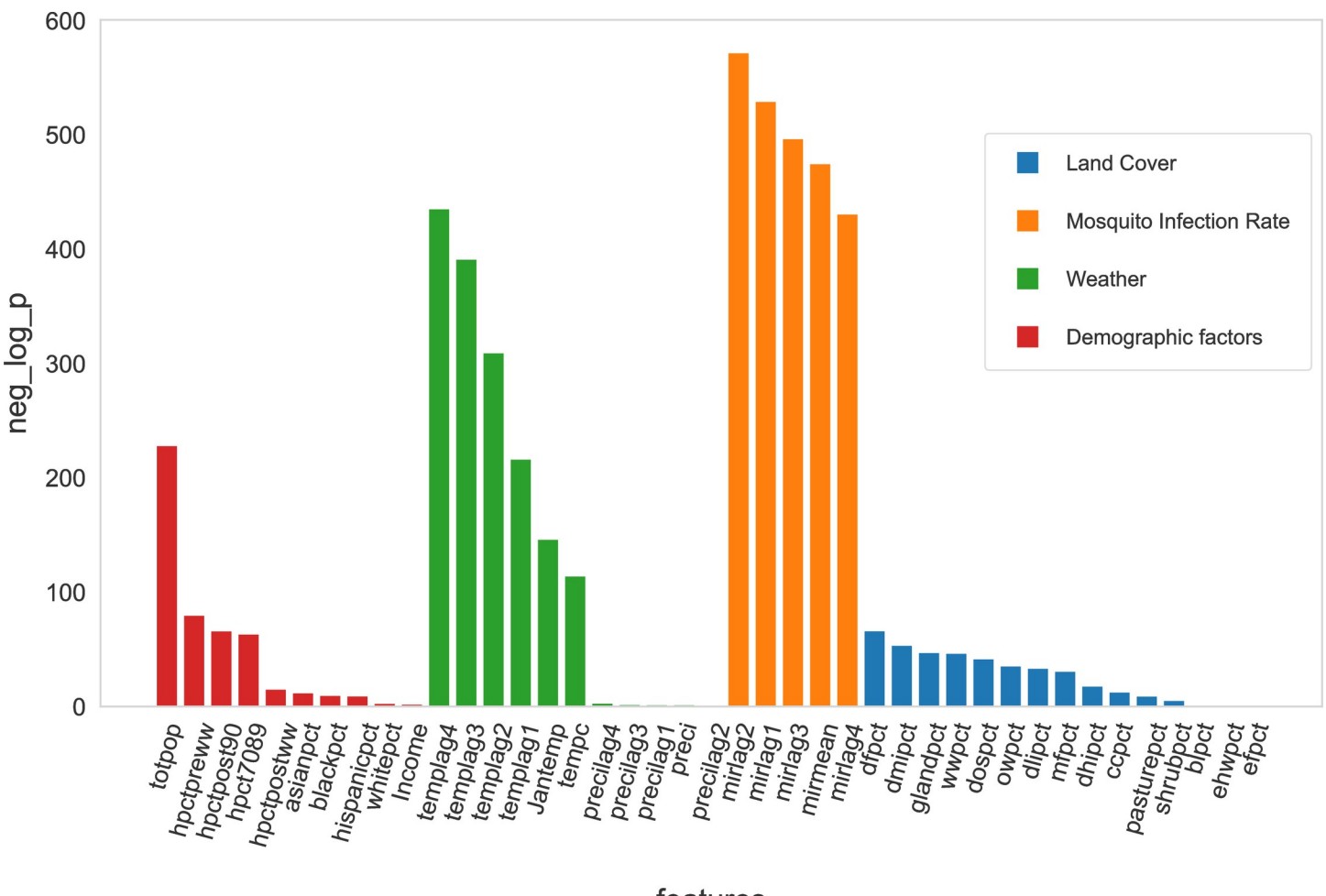

**Fig 1. -log(p) of Kolmogorov-Smirnov test for all the features and covariates.** From the KS test, we calculate the p-value, which indicates how different the distribution of the variable is between the hexagon-weeks with and without a case. The larger the -log(p), the less similar the two distributions are. The variables are grouped into four main categories. Blue bars represent the land cover variables. Orange bars represent the mosquito infection rates. Green bars represent the weather variables. Red bars represent the demographic variables.

Fig 3 shows that demographics, weather, and mosquito infection factors are candidates for strong predictors. Precipitation variables have relatively low importance among the weather factors, but still have a medium rank among the feature importance. Total population and income level, the two independent demographic variables included in the model, both have high importance in predicting WNV case occurrence. Percentage of housing built after World War II and percentage of low development intensity area are the only strong indicators among the land cover features. We ranked the features by their mean Gini feature importance, from high to low. Then, we performed t-test between each neighboring features, and found the 17th feature (owpct) is significantly lower than the 16th feature (hpctpostww). Therefore, we keep the first 16 features for a reduced model.

The cutoff for selecting the features is chosen to maximize the difference between the true positive rate (TPR) and the false positive rate (FPR). Table 3 shows the confusion matrix of the result based on the test set. With the cutoff = 0.5625, we obtain a true positive rate (recall or sensitivity) close to 0.93. The precision is about 0.006. This value is not good, but it is still well

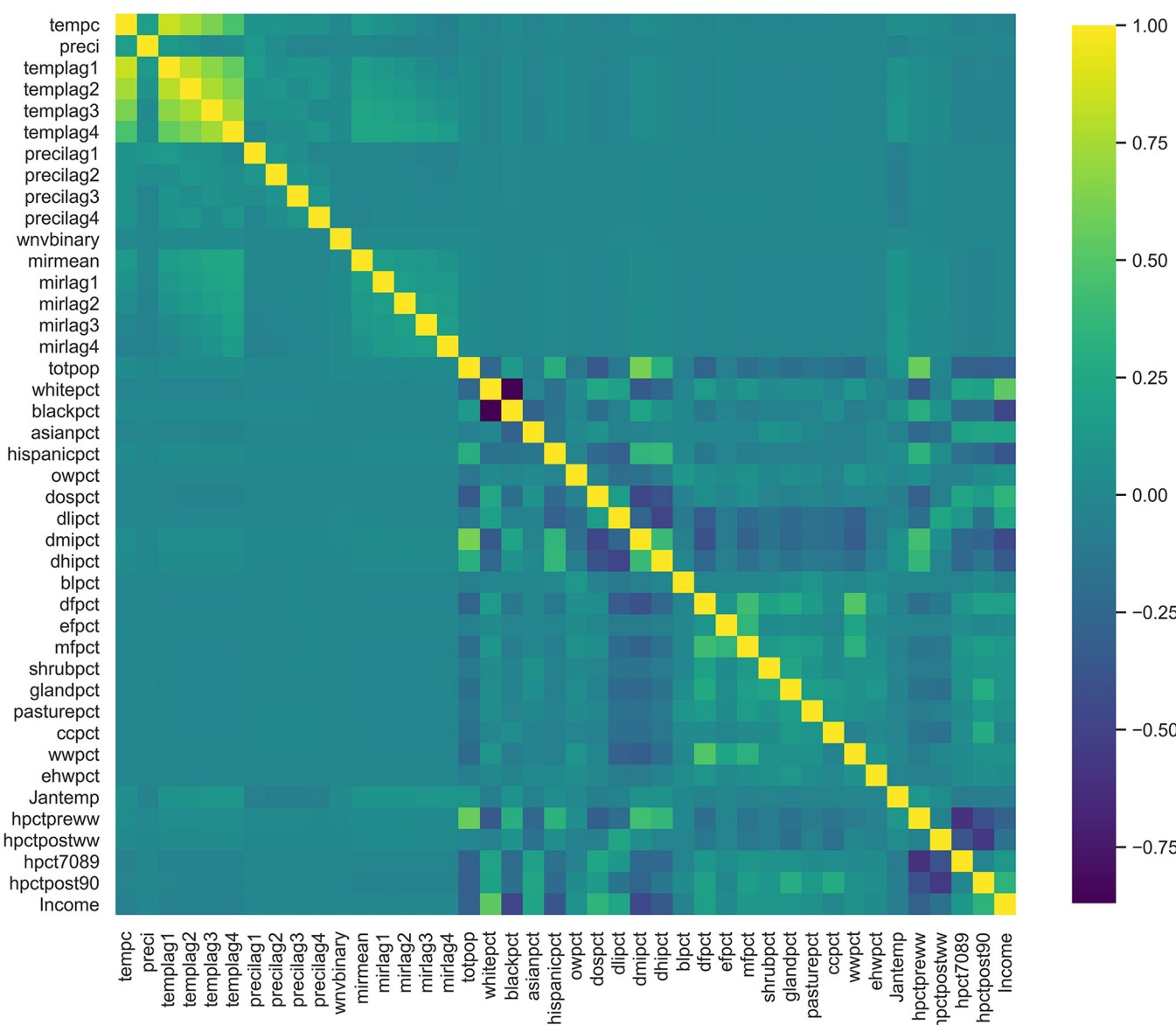

**Fig 2. Heat-map covariance matrix for all the features.** Original data are from Karki (2020) [26]. Yellow colors indicate strong positive correlations; dark blue colors indicate strong negative correlations. Light blue or green colors indicate weak correlations. We infer that temperature has a relatively high temporal correlation, as the variables tempc and templag1-4 (current temperature and temperatures 1–4 weeks before) are correlated. In addition, development stage and housing age are correlated with population, showing the interaction of population aggregation with land cover and housing status.

above the baseline derived from the proportion of positive categories (0.0005) in the dataset. The macro F1 score is 0.482 and the accuracy is 0.90. Since our model focuses on maximizing recall, this loss in overall performance is to be expected.

We predict the probability that a case of WNV will occur during a given week in each 1-km-wide hexagonal region in Cook and DuPage counties, from which we predict whether a case will occur. The receiver operating characteristic (ROC) area under the curve (AUC) is 0.96. The model has an accuracy of 0.90, a precision of 0.006, a recall of 0.92, and a macro F1 score of 0.482.

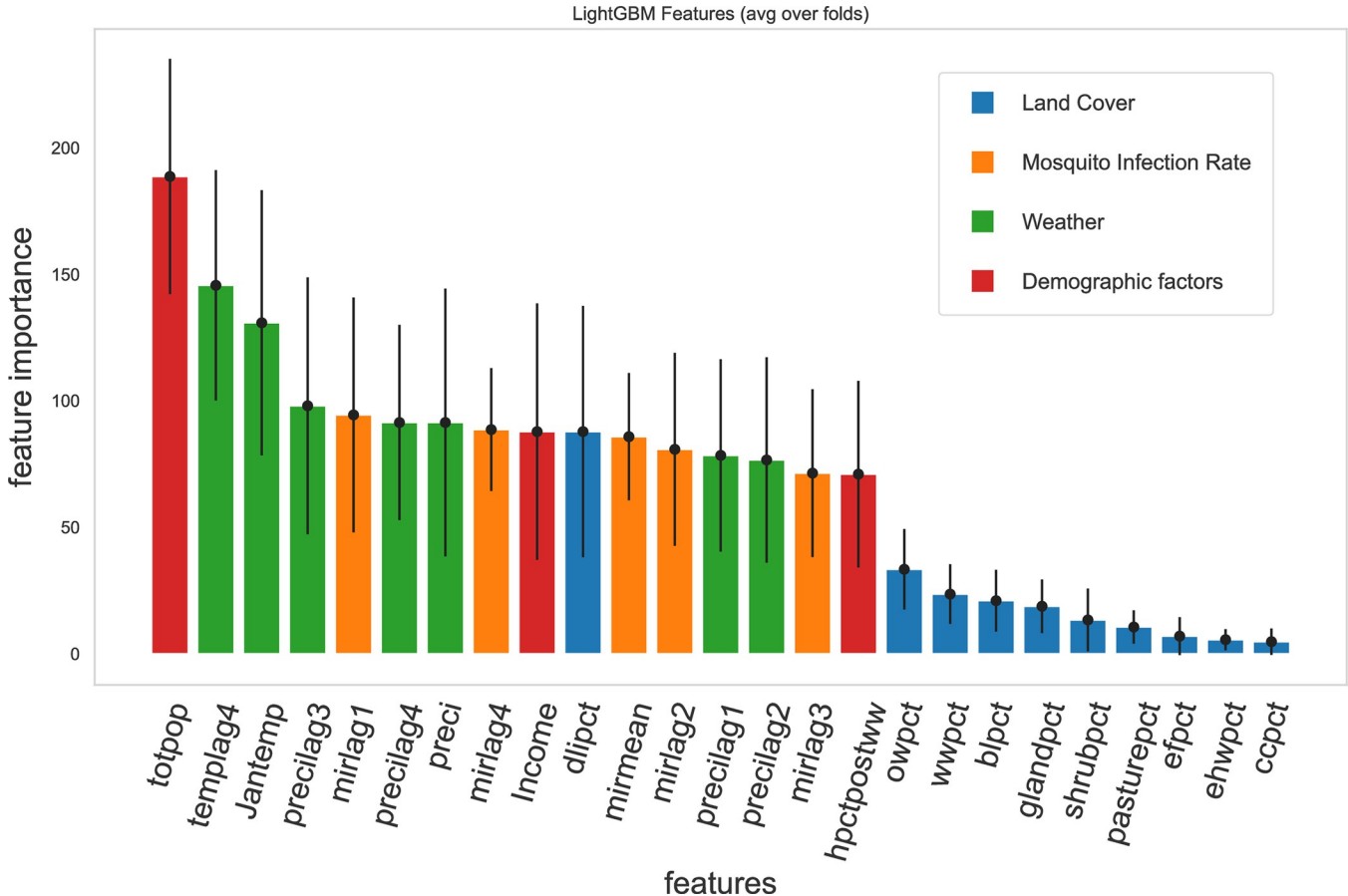

**Fig 3. Gini feature importance of the model predicting West Nile Virus cases in the Chicago area, with the 25 variables after removing the highly correlated ones.** The higher the y-value, the more important the feature is to the model. The variables are grouped into four main categories. Blue bars represent the land cover variables. Orange bars represent the mosquito infection rates. Green bars represent the weather variables. Red bars represent the demographic variables. We found that total population is the most important variable in the model. The weather and MIRs are also strong predictors.

## Light GBM model based on reduced features

We re-fit the model using only the top 16 features. The feature importance of each predictor in this model is shown in Fig 4, and its performance on the test set is shown in Table 4. The best model fitted has min_child_samples = 42, min_child_weight = 5, n_estimators = 1000, num_-leaves = 15, reg_alpha = 1, reg_lambda = 5, and subsample = 0.9373.

The cutoff for selecting the features is chosen to maximize the difference between the true positive rate (TPR) and the false positive rate (FPR). Table 4 shows the confusion matrix of the result based on the test set. With the cutoff = 0.446, we obtain a true positive rate (recall or sensitivity) close to 0.96. The precision is about 0.0034. This value is not good, but it is still well

**Table 3. Confusion Matrix of the model including all features.**

|  | hexagons with WNV cases predicted | hexagons with no WNV cases predicted |
|---|---|---|
| Hexagons with WNV Case Observed | 166 | 13 |
| Hexagons with no WNV Case Observed | 28796 | 26005 |

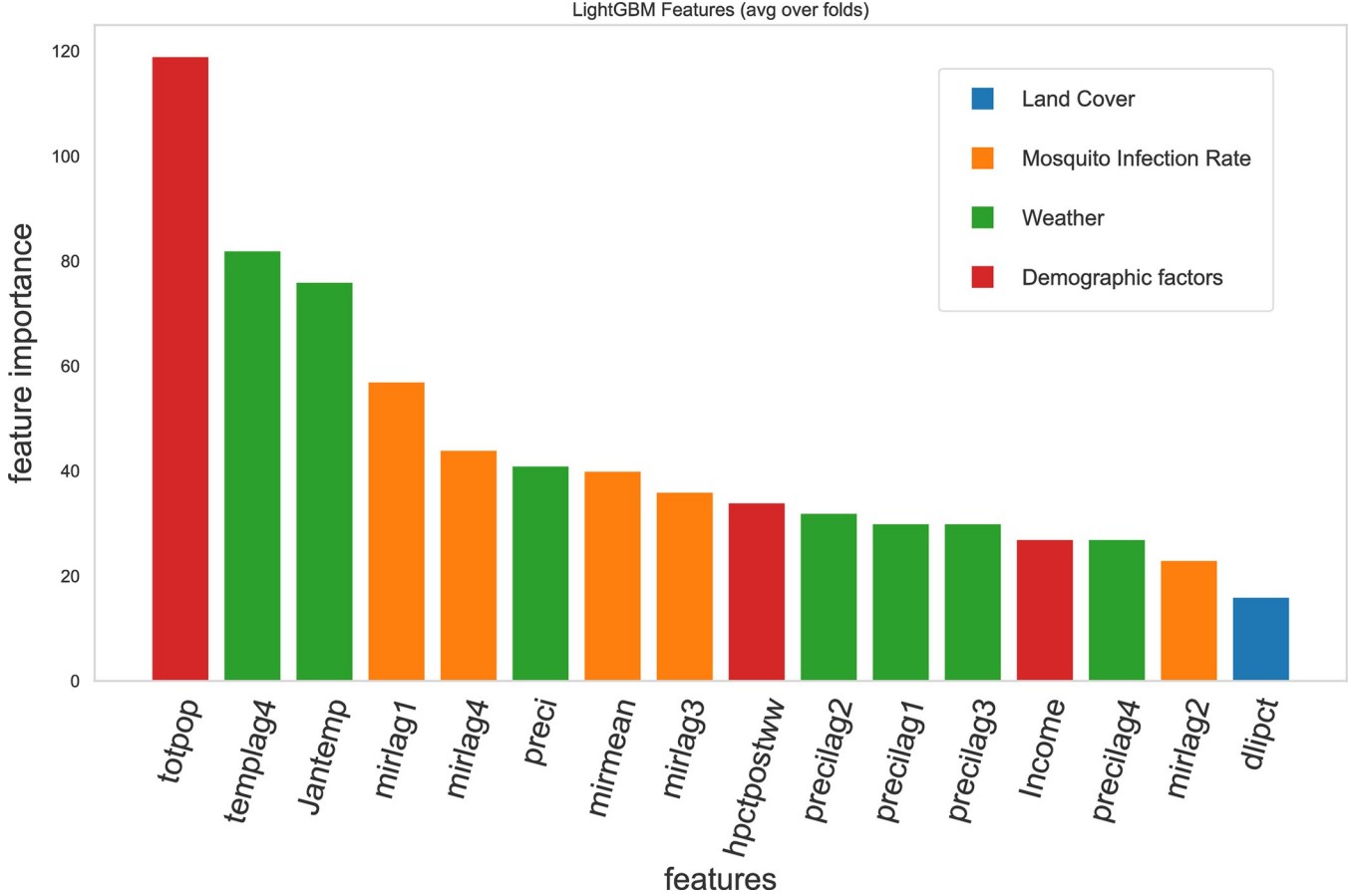

**Fig 4. Gini Feature importance of the candidate predictors in the reduced model.** The variables are grouped into four main categories. Blue bars represent the land cover variables. Orange bars represent the mosquito infection rates. Green bars represent the weather variables. Red bars represent the demographic variables. The demographic features include total population, percentage of houses built after WWII and income, ranked 1, 9, and 13. dlipct is the land cover feature selected in the model, ranking 16. The average temperature 4 weeks ago and the temperature in January are the most important weather factors. MIR 1 and 4 weeks ago are the most important MIR features. While the ranks may change in individual runs, the feature importance of these factors are close to each other.

above the baseline derived from the proportion of positive categories (0.0005) in the dataset. The F1 score is 0.45 and the accuracy is 0.83. Since our model focuses on maximizing recall, this loss in overall performance is to be expected.

We predict the probability that a case of WNV will occur during a given week in each 1-km-wide hexagonal region in Cook and DuPage counties, from which we predict whether a case will occur. The receiver operating characteristic (ROC) area under the curve (AUC) is 0.95. The model has an accuracy of 0.8267, a precision of 0.0034, a recall of 0.9664, and a macro F1 score of 0.45.

**Table 4. Confusion matrix of the reduced model.**

|  | hexagons with WNV cases predicted | hexagons with no WNV cases predicted |
| --- | --- | --- |
| Hexagons with WNV Case Observed | 173 | 6 |
| Hexagons with no WNV Case Observed | 50,060 | 238,741 |

Based on the above results, we found that the metrics of the reduced model performs similar to the model including all 25 low-correlation variables by accuracy and macro F1 score. However, when we compare the models to Karki, et.al. [24], we found the reduced model performs worse than the linear models. Therefore, we keep the full model to predict WNV incidences.

## Marginal effects

We examined the marginal effects of all the features by generating partial dependence plots. The slope of the plots shows how much each feature contributes to the model when controlling for the other factors.

Fig 5 shows the partial dependence plot of the factors that predict higher WNV risk as the values of the factors increase. MIR and total population have strong monotonic positive effects. The result is consistent that both disease-carrying mosquitoes and the human population increase the risk of infection. Weekly mean temperature 4 weeks before WNV cases are reported has a strong monotonic positive effect. It is noteworthy that the temperature range is below 30˚C, which is approximately the range that promotes mosquito activity and virus replication. January temperature also has amonotonic positive effect. One possible explanation is that, a warmer January allows mosquitoes or eggs to survive the winter, resulting in larger mosquito populations [12, 14].

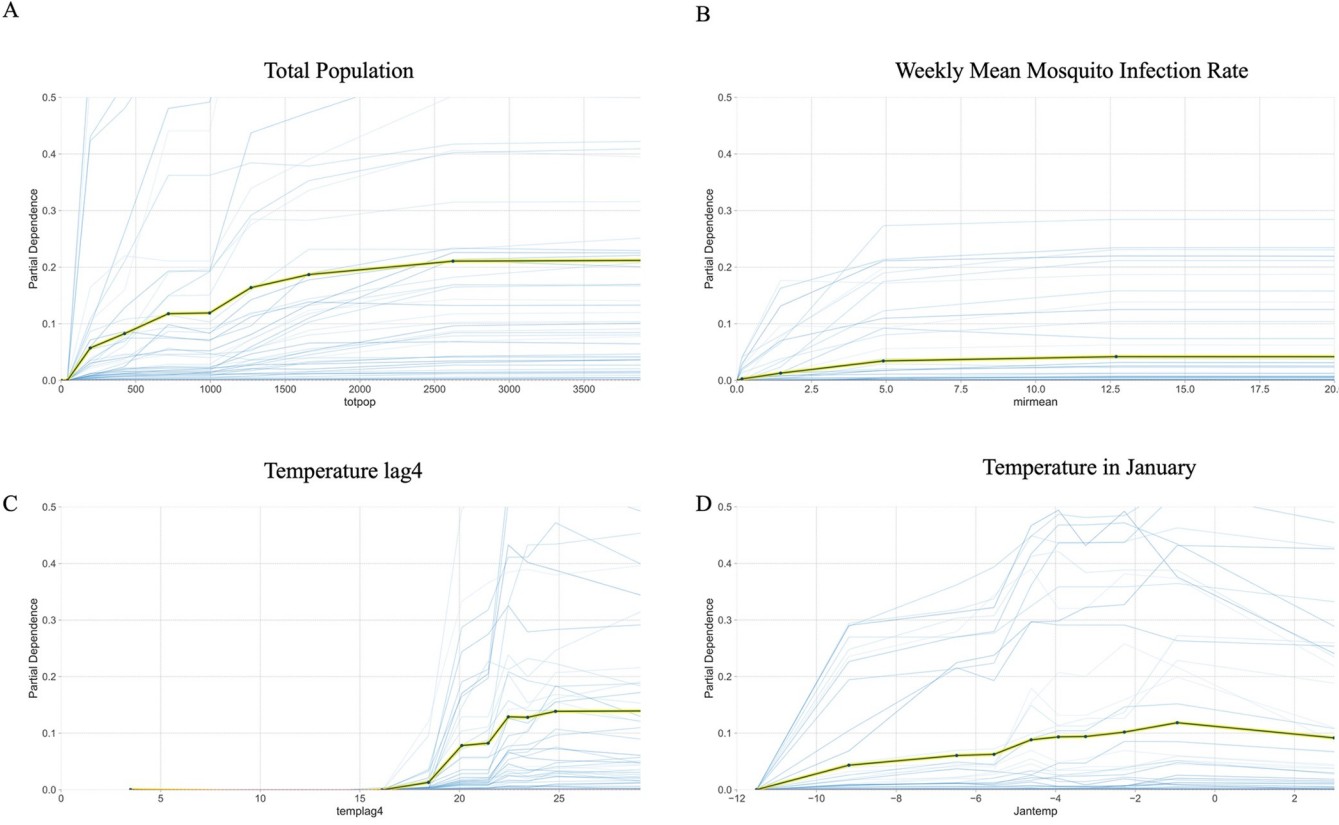

**Fig 5. Partial dependence plot of factors with positive effects: total population, mean MIR, temperature 4 weeks before WNV cases are reported, and January temperature.** The central black line is the partial dependence line, which is the average marginal effect of each factor on the WNV cases. The green shade around it is the standard deviation of the individual conditional expectation (ICE) lines, which is the predicted marginal effect by each sample of each factor on the WNV cases. The blue shades are samples from the ICE lines, showing the range of predicted marginal effects by each individual sample. The MIR and the weekly temperatures in 1–4 weeks before also have similar trends as the mean MIR and the temperature of the current week.

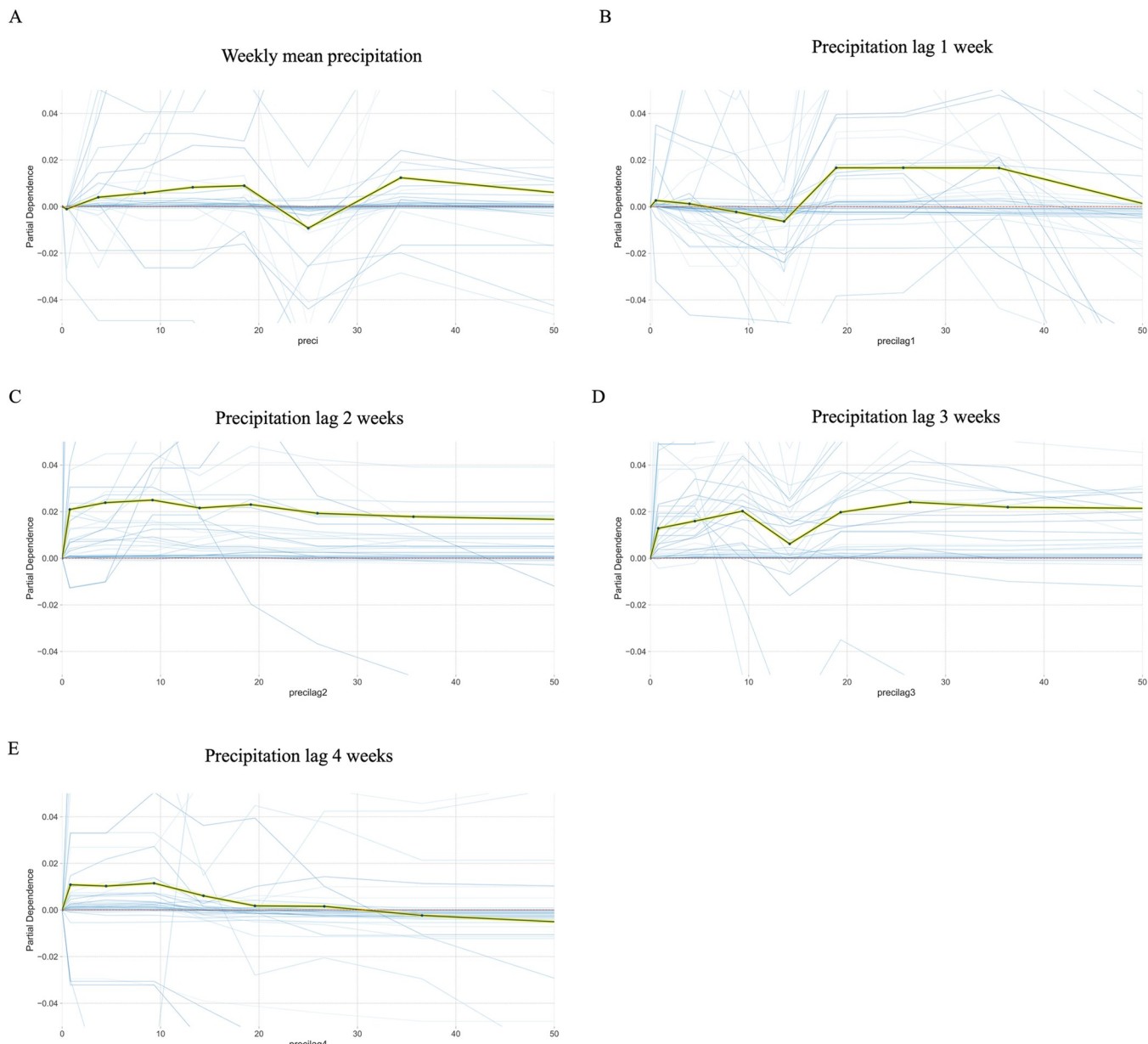

**Fig 6. Partial dependence plot of precipitation for the current week and 1–4 weeks prior.** The central black line is the partial dependence line. The green shade around it is the standard deviation of the ICE lines. The blue shades are samples from the ICE lines. Precipitation variables have non-monotonic effects.

On the other hand, as shown in Fig 6, the precipitation variables have non-monotonic effects. This result is consistent with the existing literature [10, 14, 26]. The effect of precipitation is inconsistent and complex, that there is no monotonic effect. While temporary water accumulation provides mosquitoes with more places to lay eggs, excessive precipitation can also wash away mosquito eggs, thus reducing the risk of WNV. However, the weather water will accumulate or wash away the mosquito eggs depends on the types and formation of land surface, and therefore we cannot see a definite trend in the precipitations.

As shown in Fig 7, apart from total population (in Fig 1), land cover and socioeconomic features have relatively small effects. We don't observe a strong marginal effect of income,

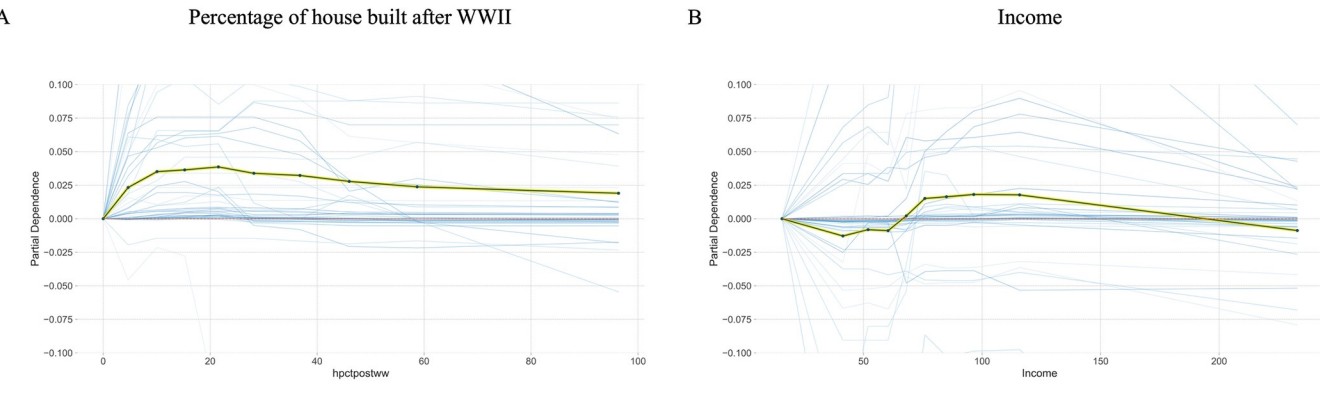

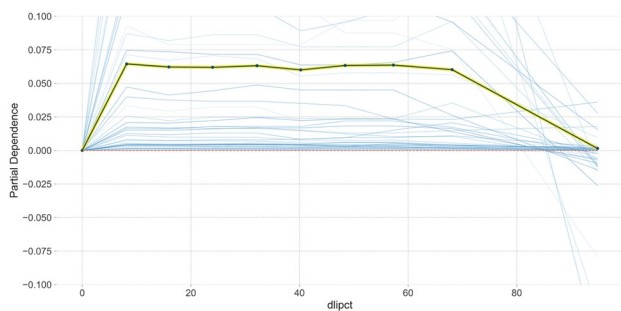

**Fig 7. Partial dependence plot of socioeconomics and land cover features.** The central black line is the partial dependence line. The green shade around it is the standard deviation of the ICE lines. The blue shades are samples from the ICE lines. The socioeconomics and land cover features are not very strongly represented. There is not a very strong marginal effect of income. The percentage of houses built after World War II has a slight negative effect, indicating that people living in older neighborhoods have higher WNV risks. Meanwhile, the percentage of less developed land has a slight positive effect at the lower end.

although it is presented in the feature selection. House age and land development intensity both have small effects on WNV case prediction.

## Conclusion

We performed two-step light GBM procedures to identify a minimum model. We evaluated the ROC-AUC score, accuracy, recall, precision and F-1 score of the models. We found that the reduced model has a worse performance than the linear models of Karki, et. al. [26], while the full model has a similar performance. Therefore, we kept all 25 parameters in the model for prediction. We have found that the natural effects including January temperature, weekly temperature (lagged 0–4 weeks), weekly precipitation (lagged 0–4 weeks), and weekly MIR (lagged 0–4 weeks), as well as the total population are the dominant features that are strongly correlated with the incidence of West Nile virus human cases.

We found consistent features with Karki, et al. that mosquito infection rate, temperature and their lag effects are important factors [26]. The mosquito infection rate, temperatures, and their lag effects have high feature importance to predict the WNV incidences. This result was further confirmed with PD plots. The mosquito infection rate, temperatures, and their lag effects all show positive marginal effects. These effects are also identified as strong predictors in the linear models by Karki, et al [26]. We also found the behavior of precipitation factors

consistent with the literature [10, 14, 26], being median predictors with non-monotonic marginal effects. Keyel et.al. found the effect of precipitation complex and not consistently detected [10], while our results show the marginal effect of precipitation is small compared to mosquito infection rate and temperature. Moreover, precipitation does not have a monotonic effect, indicating it is complex. Poh et.al. stated that, while precipitation have important effect on mosquito productivity and abundance, its pattern to influence WNV incident is complex and unclear [14]. Karki et al. also finds precipitation variables in their linear models, with low effects. We believe the low effects in the linear models result from the non-monotonic effect.

In addition, we found that the percentage of houses built after World War II, which is not included in the original work, is quite important. While income is selected as a predictor by the final model, the PD plot has shown that it has very low marginal effects. Moreover, we have expected different types of land cover might affect mosquito reproduce site by their abilities to keep water on the surface, but did not uncover as strong effect as the weather and mosquito infection rates. In addition to non-monotonicity, one possible explanation is that both the number of cases, the weather, and the mosquito infection rates are measured on a weekly basis, while the land cover and socioeconomic data are static. Therefore, the land cover and socioeconomic features have less variations, and their effects are harder to uncover.

One concern was that the behavior of the model may differ by the income of the area, as income disparities may affect diagnosis rates, surveillance efforts, and distribution of land cover and housing variables. Therefore, the light GBM model fitting was repeated for subsets of the data consisting of the areas with above-median income and the areas with below-median income (S3 File). These stratified models were similar to each other and to the full model, indicating that the predictive capabilities of this model are not predicated on income groupings.

In conclusion, our light GBM model provides an alternative way to predict the probability of an area having a WNV case or not. The performance in terms of ROC-AUC is very close to the previous work [26] and is much better at detecting the area where there is actually a case. We also have a clearer relationship between temperature and precipitation, mosquito infection, and West Nile virus. In addition, we identified weak effects of socioeconomics and land cover. The risk of contracting WNV does not appear to be related to income in these data. However, other factors may relate to income and WNV detection that are not possible to study with these data, such as variation in diagnosis rates.

The results of this study can be used as a guideline to develop a threshold for public health intervention.

## Supporting information

**S1 File. Socioeconomic data cleaning and transformation.** In this section, we explained the details in preprocessing the 5-year rolling average of socioeconomic data in the years 2010–2017.
(DOCX)

**S2 File. Modeling with socioeconomic data with 5-year rolling average 2010–2017.** In this section compares the socioeconomic data we obtained to the static ones Karki, et.al. used [26] and found the results are similar. Therefore, we used the Karki data.
(DOCX)

**S3 File. Models stratified by income.** In this section, we retrained our model with the data stratified by income higher than or equal to the medium income. We have found the high/low-income models are similar.
(DOCX)

**S4 File. KS importance table.** The -log($p$-values) obtained from the KS test are listed for each factor. For a group of factors that has correlation larger than 0.35, we will only keep the one with the highest -log($p$-values).
(XLSX)

## Acknowledgments

The authors would like to thank the HAL cluster and support team for providing the computational resources to complete the work. The author would also like to acknowledge the efforts of the NCSA Industry Group for supporting the work. The authors would like to thank Dr. Christina Fliege for her editorial suggestions on this manuscript. The authors would like to thank Mr. Mingyu Yang for his help in retrieving and preprocessing the census data.

## Author Contributions

**Conceptualization:** Guangya Wan, John Uelmen, Liudmila Sergeevna Mainzer, Rebecca Lee Smith.

**Data curation:** Guangya Wan, Shubham Rawlani, John Uelmen, Rebecca Lee Smith.

**Formal analysis:** Guangya Wan, John Uelmen, Rebecca Lee Smith.

**Funding acquisition:** Liudmila Sergeevna Mainzer.

**Investigation:** Liudmila Sergeevna Mainzer, Rebecca Lee Smith.

**Methodology:** Guangya Wan, Joshua Allen, Shubham Rawlani, John Uelmen.

**Software:** Guangya Wan.

**Supervision:** Joshua Allen, Weihao Ge, John Uelmen, Liudmila Sergeevna Mainzer, Rebecca Lee Smith.

**Validation:** Guangya Wan, John Uelmen.

**Visualization:** Guangya Wan.

**Writing – original draft:** Guangya Wan.

**Writing – review & editing:** Joshua Allen, Weihao Ge, John Uelmen, Liudmila Sergeevna Mainzer, Rebecca Lee Smith.

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
