## [Decision Letter · Decision Letter 0]

18 Sep 2023

PONE-D-23-17817Two-Step Light Gradient Boosted Model to identify human West Nile Virus infection risk factor in ChicagoPLOS ONE

Dear Dr. Ge,

Thank you for submitting your manuscript to PLOS ONE. After careful consideration, we feel that it has merit but does not fully meet PLOS ONE’s publication criteria as it currently stands. Therefore, we invite you to submit a revised version of the manuscript that addresses the points raised during the review process.

We look forward to receiving your revised manuscript.

Kind regards,

José Ramos-Castañeda, M.Sc., Ph.D

Academic Editor

PLOS ONE

Journal Requirements:

Additional Editor Comments:

I have read the manuscript in detail and consider that this manuscript should be extensively reviewed and revised. One of the reviewers has mentioned in detail aspects that should be considered in the revised manuscript.

Reviewers' comments:

Reviewer's Responses to Questions

**Comments to the Author**

1. Is the manuscript technically sound, and do the data support the conclusions?

Reviewer #1: Yes

Reviewer #2: Partly

2. Has the statistical analysis been performed appropriately and rigorously? 

Reviewer #1: Yes

Reviewer #2: Yes

3. Have the authors made all data underlying the findings in their manuscript fully available?

Reviewer #1: Yes

Reviewer #2: Yes

4. Is the manuscript presented in an intelligible fashion and written in standard English?

Reviewer #1: Yes

Reviewer #2: Yes

5. Review Comments to the Author

Reviewer #1: The author is requested to include quantitative values within the conclusions. Thinking of concluding based on the initial objectives of the project. On the other hand, it is requested that the graphics be inserted in a more readable format, the images are pixelated. Please improve the quality of the figures. Within the text figure 6 remains to be detailed. Regarding the utility of the program as a predictive model, it is a very valuable contribution.

Reviewer #2: Wan et al. performed a two-step modeling approach to predict WNV in the Chicago area and identify important parameters for prediction. Work to improve WNV prediction is vital in protecting human health so this work could be a notable addition. Their approach appears to build off a previous model by Karki et al., but the motivation for this is not well defined nor is the improvement obtained by this model over the previous results. Also, the methods and results are often difficult to follow and only a cursory discussion is made of the results. The data selection and model building procedures could benefit from clarification (e.g., rewording, visual diagram of the steps in the process). I have several comments and questions that need to be addressed to improve the understanding and potential utility of the results for WNV prediction.

Line 29-59. Please consider re-structing or breaking up this single paragraph into smaller paragraphs to aid the reader in following the multiple topics included. As I see it, this paragraph mentions previous predictive models as well as the relationship of climate, land cover, socioeconomic factors with WNV infection in mosquitoes and humans in a list-like fashion (i.e., this author found this list of covariates as significant and this other author found this list of covariates significant). Previously developed predictive models are then re-visited in more depth in the following paragraph.

Line 35. The statement is incorrect. Hahn et al. found a nonsignificant relationship between above-average temperatures and WNV risk in the western regions.

Line 48. The term “geologic” might be used incorrectly when referring to land cover and types of land use.

Line 51. Please provide more explanation on how ethnicity is a key predictor of WNV circulation in humans as there is no direct causal relationship between ethnicity (or race) and disease incidence. You indicate in Line 59 that socioeconomic differences between racial groups explained differences in prevalence.

Line 68. Please provide more information in light GBM models. The current description would describe random forests as well, but the following sentence indicates that there are differences in fitting speed. How are light GBM and random forest models different?

Line 75. Do you mean socioeconomics instead of social economics?

Line 83. Assuming the data (WNV case, mosquito surveillance) are available for the subsequent years, could you use your model trained on 2005-216 to predict more recent years as an external validation of model performance?

Line 93. Why was weather data aggregated from the native grid to the census tract and then to the hexagonal grid? Why not aggregate directly from the original grid to the hexagonal grid without the potential loss in information across multiple aggregation steps of different spatial scales?

Line 109 (Table 1). Were all MIR variables the mean during the time period (0-4 weeks) or just at the 0-week lag? Similarly, were the temperature variables mean, min, or max during the time periods? Also, the “nature” category seems a bit odd to me. It seems more appropriate/understandable to break it into two categories: mosquito surveillance and weather.

Line 114. This statement is confusing. Do you mean that the KS test is not designed to test for collinearity among predictors?

Line 122. The statement “therefore, it is possible that the factors selected in the model are correlated with the true predictors” is confusing. What do you mean by “true predictors”?

Line 124. What metric did you use to identify functional significance of covariates for inclusion in the model?

Line 128. Please provide more details on the model building and validation. The Results (Line 190) indicate that cross-validation was performed, but no details are given here in the Methods. Similarly, what hyperparameters were selected from the grid search?

Line 155. The KS test is not designed to assess causality. Please rephrase your statement to something like the following “characteristics were not significantly related to WNV risk”.

Line 159, 195. The mention of the four main + one residual categories in the figure legends is confusing as only three categories have been mentioned in the text and Table 1.

Line 181. Please list what the variables with the highest KS scores were. These variables could be gleaned from the figures, but it is unclear in the text which of the medium to highly correlated variables were included in the model.

Line 191. Please indicate which metric was used to discern variable importance as multiple metrics could have been used. This is stated in the legend for Fig 3, but not in the text itself.

Line 206. Consider re-wording to “We observed a natural break in variable importance (approx.. 20) between MIR at 3-week lags (mirlag3) and percentage of open water (owpct)” to clarify the meaning and align with the statement in Line 222.

Lines 209 and 230. Clarify the feature selection resulting from maximizing TPR and FPR in the test set. It is not clear if any features were removed during this step.

Line 215. It appears that the results in Table 2 are from the full model (Line 210) as well as from the reduced model (Line 223). Please clarify which model was used and provide further results for the other model’s performance on the test dataset.

Line 229. Given variation in the runs, it could be prudent to provide some indication in the plots (Figs 3-4) on the magnitude of variation (e.g., 95% CI or ranges).

Line 247. Were partial dependence plots produced for all covariates in the full or reduced model?

Line 249 (Figs 5-7). Does the shaded region represent a 95% CI of the marginal effects? If so, what is the significance of this interval containing 0 for most of the covariates? Also, please provide units for covariates (x-axis) and marginal effect (y-axis).

Line 259. Please provide support for your supposition that warmer January temperatures increase mosquito survival over the winter.

Line 281. This conclusion appears to contradict your statement in Line 243 that the reduced model was sufficient to describe the results.

Line 285. To what are you comparing the recall of the light GBM model? Additionally, how does this statement relate to the discussion of importance features that follows?

Line 288. Please expound on what these consistent relationships are instead of just citing references and leaving it up to the reader to deduce what these relationships are.

Line 290. Why might income have been identified as an important variable and yet have no marginal effect?

Line 295. Given the effort taken to reduce collinearity among predictors, how much correlation remained between lagged MIR and lagged temperature that could mask the relationship of socioeconomic factors and WNV incidence?

Line 309. Given the outlined conclusions, I do not see how the results of this model enhance the ability to predict WNV incidence. Please provide more details on how this study could develop or modify currently existing thresholds for public health.

6. PLOS authors have the option to publish the peer review history of their article (what does this mean?). If published, this will include your full peer review and any attached files.

Reviewer #1: No

Reviewer #2: No

---

## [Author Response · Author response to Decision Letter 0]

18 Nov 2023

Reviewer #1: The author is requested to include quantitative values ??within the conclusions. Thinking of concluding based on the initial objectives of the project. On the other hand, it is requested that the graphics be inserted in a more readable format, the images are pixelated. Please improve the quality of the figures. Within the text figure 6 remains to be detailed. Regarding the utility of the program as a predictive model, it is a very valuable contribution.

Reviewer #2: Wan et al. performed a two-step modeling approach to predict WNV in the Chicago area and identify important parameters for prediction. Work to improve WNV prediction is vital in protecting human health so this work could be a notable addition. Their approach appears to build off a previous model by Karki et al., but the motivation for this is not well defined nor is the improvement obtained by this model over the previous results. Also, the methods and results are often difficult to follow and only a cursory discussion is made of the results. The data selection and model building procedures could benefit from clarification (e.g., rewording, visual diagram of the steps in the process). I have several comments and questions that need to be addressed to improve the understanding and potential utility of the results for WNV prediction.

Line 29-59. Please consider re-structing or breaking up this single paragraph into smaller paragraphs to aid the reader in following the multiple topics included. As I see it, this paragraph mentions previous predictive models as well as the relationship of climate, land cover, socioeconomic factors with WNV infection in mosquitoes and humans in a list-like fashion (i.e., this author found this list of covariates as significant and this other author found this list of covariates significant). Previously developed predictive models are then re-visited in more depth in the following paragraph.

Sure. We are breaking up the paragraph and modifying the texts. In this paragraph, our goal is to list factors influencing human incidences from the literature. In the literature we found the weather, land cover, and socioeconomic factors. The impacts of these factors on infection in mosquitoes are also considered. The models, mainly linear regression and tree methods, are discussed in the next paragraph.

Line 35. The statement is incorrect. Hahn et al. found a nonsignificant relationship between above-average temperatures and WNV risk in the western regions.

Thank you. We have double-checked the paper by Hahn et. al. and modified the statement.

Line 48. The term “geologic” might be used incorrectly when referring to land cover and types of land use.

Thank you. The term is corrected to “land cover”.

Line 51. Please provide more explanation on how ethnicity is a key predictor of WNV circulation in humans as there is no direct causal relationship between ethnicity (or race) and disease incidence. You indicate in Line 59 that socioeconomic differences between racial groups explained differences in prevalence.

According to Hernandez et al., the racial disparity of the disease incidence might be due to underreport affected by health insurance and willingness to seek medical care. Moreover, there are behavioral risk associated with different ethnic groups, like whether the individuals working outside home.

Line 68. Please provide more information in light GBM models. The current description would describe random forests as well, but the following sentence indicates that there are differences in fitting speed. How are light GBM and random forest models different?

Yes, we agree that our previous description is not accurate for the lightGBM model, which is not an ensemble tree method. The lightGBM model is similar to random forest in that both models select features based on decision trees. However, LightGBM, as its name indicated, is a gradient boosting algorithm, that the decision trees are trained based on the errors of the previous trees. On the other hand, random forest is an ensemble method tha trains multiple decision trees independently, and then combines the predictions of the trees via averaging or voting. The random forest used a depth-wise strategy for growing trees while lightGBM used a leaf-wise strategy. Therefore, the trees by random forest are more balanced, and the lightGBM approach converges faster. 

Line 75. Do you mean socioeconomics instead of social economics?

Yes, we corrected “social economics” to “socioeconomics”

Line 83. Assuming the data (WNV case, mosquito surveillance) are available for the subsequent years, could you use your model trained on 2005-216 to predict more recent years as an external validation of model performance?

Yes, our model can also be used to train and predict more recent years. The WNV case number can be obtained from local health department. However, the weekly mosquito infection rate at the 1km hexagons needs to be collected.

Line 93. Why was weather data aggregated from the native grid to the census tract and then to the hexagonal grid? Why not aggregate directly from the original grid to the hexagonal grid without the potential loss in information across multiple aggregation steps of different spatial scales?

Yes, we checked back with Karki, et.al. and found we have made a mistake. The weather data are aggregated to the hexagonal grids directly from the 4km resolution PRISM data. The socioeconomics data are downloaded at the census tract level from the census beaureau.

Line 109 (Table 1). Were all MIR variables the mean during the time period (0-4 weeks) or just at the 0-week lag? Similarly, were the temperature variables mean, min, or max during the time periods? Also, the “nature” category seems a bit odd to me. It seems more appropriate/understandable to break it into two categories: mosquito surveillance and weather.

We have modified Table 1, and make the general categories same as the Table 1 in Karki. et.al., that is: Land cover, Mosquito infection rate, Weather, and Demographic factors. MIRmean, MIRlag1-4, preci, precilag1-4, tempc, templag1-4 are weekly means. 

Line 114. This statement is confusing. Do you mean that the KS test is not designed to test for collinearity among predictors?

Yes, KS test is not to test the collinearity and correlation between the predictors, therefore, we calculated the Pearson’s correlation and filtered out more features before we started modeling.

Line 122. The statement “therefore, it is possible that the factors selected in the model are correlated with the true predictors” is confusing. What do you mean by “true predictors”?

By “true predictors”, we wanted to say that the strongest features identified by the model are not necessarily the causal factors for the WNV.

Line 124. What metric did you use to identify functional significance of covariates for inclusion in the model?

We used Pearson’s correlation to group the covariates. Then, in each group, we selected the ones with the highest -log(p-value) in the KS test.

Line 128. Please provide more details on the model building and validation. The Results (Line 190) indicate that cross-validation was performed, but no details are given here in the Methods. Similarly, what hyperparameters were selected from the grid search?

We used a randomized search with a 5-fold cross validation. We added Table 2 for the hyperparameter range we are searching. 

Line 155. The KS test is not designed to assess causality. Please rephrase your statement to something like the following “characteristics were not significantly related to WNV risk”.

Thank you. We rephrased the statements.

Line 159, 195. The mention of the four main + one residual categories in the figure legends is confusing as only three categories have been mentioned in the text and Table 1.

The figures are regenerated to match the categories to the ones in Table 1.

Line 181. Please list what the variables with the highest KS scores were. These variables could be gleaned from the figures, but it is unclear in the text which of the medium to highly correlated variables were included in the model.

We include KS results in supplementary material 4 now.

Line 191. Please indicate which metric was used to discern variable importance as multiple metrics could have been used. This is stated in the legend for Fig 3, but not in the text itself.

We used the Gini feature importance to discern variable importance.

Line 206. Consider re-wording to “We observed a natural break in variable importance (approx.. 20) between MIR at 3-week lags (mirlag3) and percentage of open water (owpct)” to clarify the meaning and align with the statement in Line 222.

Thank you for the suggestion. We re-run the lightGBM analysis for 5 times, took the 6 data sets, calculated the error values and means of each feature. Then we ranked the features by feature importance from high to low. We performed t-test between each neighboring features and found at rank 16, the features are significantly different from each other (p=0.03)

Lines 209 and 230. Clarify the feature selection resulting from maximizing TPR and FPR in the test set. It is not clear if any features were removed during this step.

Thank you for pointing this out. We wrongly stated the threshold is for feature selection. But the threshold is to determine whether a WNV case occurs in a hexagon grid in a week. No features were removed during this step. 

With the models, we predict the probabilities of a WNV case occurrence in a 1km hexagon grid during a week. If the probability is greater than equal to the threshold, we define the WNV case occurs. If the probability is less than the threshold, we define the WNV case doesn’t occur. Then the program checks the prediction with the data set to see if they match, and categorized the result in to true positive, true negative, false positive, and false negative. Then the difference of TPR and FPR are calculated for each threshold. The threshold for the largest difference of TPR and FPR are selected to determine if a hexagon grid has WNV case occurrence.

Line 215. It appears that the results in Table 2 are from the full model (Line 210) as well as from the reduced model (Line 223). Please clarify which model was used and provide further results for the other model’s performance on the test dataset.

Thank you. We have corrected the tables and the text. The “table 2” (new table 3) is the confusion matrix for the full model at the best threshold for predicted probability, and the “table 3” (new table 4) is the reduced model at the best threshold for predicted probability.

Line 229. Given variation in the runs, it could be prudent to provide some indication in the plots (Figs 3-4) on the magnitude of variation (e.g., 95% CI or ranges).

We have added the 95% CI lines for figure 3. However, the reduced model does not have any repetitive runs. Therefore figure 4 doesn’t have 95% CI lines.

Line 247. Were partial dependence plots produced for all covariates in the full or reduced model?

The partial dependence plots are produced for the full model

Line 249 (Figs 5-7). Does the shaded region represent a 95% CI of the marginal effects? If so, what is the significance of this interval containing 0 for most of the covariates? Also, please provide units for covariates (x-axis) and marginal effect (y-axis).

Thank you for pointing this out. Yes, we have made a mistake in generating the partial dependence plot. The shaded region represents the lines of all predicted outcome lines. We have changed the settings and produced the figures again. The yellow shade around the center line is the variability of the central tendency. The extra lines are the predicted outcome lines.

Since the data is mainly WNV_binary == 0, therefore there will always been predicted outcome lines lying on the y==0 place, if the data sampled to produce the lines all have WNV_binary==0. Therefore, if we plot the full range, it will always pass 0s. In the new figures, the central yellow shade shows the central tendency, and the trend is clearer.

Line 259. Please provide support for your supposition that warmer January temperatures increase mosquito survival over the winter.

Hahn, et.al. has found that warmer December and January temperatures correlated with the WNV incidences in the later months. They suggested that a warmer winter will increase mosquito and egg survival in the winter. Poh, et.al. has also found warmer winter has associated with mosquito abundance as well as WNV outbreak 8 months later.

Line 281. This conclusion appears to contradict your statement in Line 243 that the reduced model was sufficient to describe the results.

Yes, we agree. We have changed the analysis to the full models.

Line 285. To what are you comparing the recall of the light GBM model? Additionally, how does this statement relate to the discussion of importance features that follows?

Yes, we agree with the comment. The linear models do not provide a recall. Therefore, we cannot compare the recall of the lightGBM models to the linear models. The sentence is removed and the whole paragraph now focuses on important feature discussion

Line 288. Please expound on what these consistent relationships are instead of just citing references and leaving it up to the reader to deduce what these relationships are.

Yes. We have added these in the discussion.;

Line 290. Why might income have been identified as an important variable and yet have no marginal effect?

Income was selected as one of the top features in our model. However, according to the new partial dependency plot, its marginal effect is very low. Its effect first decreases and then increases. We believe income disparity of WNV incidences is similar to what Hernandez et al. described for racial disparity. In the group lower than the median $65,000, it is likely that people are reluctant to seek medical care when WNV symptoms are mild, therefore having an under reporting. The slight positive effect following it might be related to working location and other behavioral risks. 

Line 295. Given the effort taken to reduce collinearity among predictors, how much correlation remained between lagged MIR and lagged temperature that could mask the relationship of socioeconomic factors and WNV incidence?

Our statement that the socioeconomic and land cover effect was masked was wrong. We aggregated the data by year and calculated the models again, and did not find any strong effect. We have to say that the socioeconomic and land cover really have weaker effect than MIR and temperature.

Line 309. Given the outlined conclusions, I do not see how the results of this model enhance the ability to predict WNV incidence. Please provide more details on how this study could develop or modify currently existing thresholds for public health.

Although the LightGBM method does not show improved predictive power over the linear model, its predictive power is similar and it makes those predictions using a different modeling process. This makes LightGBM suitable for inclusion in model ensembles, which benefit from inclusion of models built on different frameworks and underlying assumptions.

---

## [Decision Letter · Decision Letter 1]

10 Dec 2023

Two-Step Light Gradient Boosted Model to identify human West Nile Virus infection risk factor in Chicago

PONE-D-23-17817R1

Dear Dr. Ge,

We’re pleased to inform you that your manuscript has been judged scientifically suitable for publication and will be formally accepted for publication once it meets all outstanding technical requirements.

Kind regards,

José Ramos-Castañeda, M.Sc., Ph.D

Academic Editor

PLOS ONE

Additional Editor Comments (optional):

Thank you for having responded to the reviewers' comments and suggestions. One reviewer has made editorial comments on your manuscript, so the article would benefit in clarity if they were carried out. This can be done at the post-acceptance revision stage.

Reviewers' comments:

Reviewer's Responses to Questions

**Comments to the Author**

1. If the authors have adequately addressed your comments raised in a previous round of review and you feel that this manuscript is now acceptable for publication, you may indicate that here to bypass the “Comments to the Author” section, enter your conflict of interest statement in the “Confidential to Editor” section, and submit your "Accept" recommendation.

Reviewer #1: All comments have been addressed

Reviewer #2: (No Response)

2. Is the manuscript technically sound, and do the data support the conclusions?

Reviewer #1: Yes

Reviewer #2: Yes

3. Has the statistical analysis been performed appropriately and rigorously? 

Reviewer #1: Yes

Reviewer #2: Yes

4. Have the authors made all data underlying the findings in their manuscript fully available?

Reviewer #1: Yes

Reviewer #2: Yes

5. Is the manuscript presented in an intelligible fashion and written in standard English?

Reviewer #1: Yes

Reviewer #2: Yes

6. Review Comments to the Author

Reviewer #1: The author justified the observations mentioned in the review of both reviewers 1 and 2. Therefore, it is suggested that we proceed to publication.

Reviewer #2: Thank you for addressing my previous comments and suggestions. I believe that further work is needed to improve the clarity of the manuscript such that these findings could be meaningful for public health and vector control. In particular, the manuscript would benefit from another review by the authors with an eye towards grammar and removing redundant sentences. Also, the Conclusions are somewhat hard to follow and could be linked back to the original goals of the project for clarity for the readers. It is unclear how or why lightGBM models like yours could improve WNV prediction.

I illustrate a few places with redundant sentences below:

Lines 133-134. The statement “KS test is distribution-free” is redundant to “it doesn’t rely on distribution assumptions”.

Lines 145-148 are restatements of Lines 140-143.

Other comments:

Figures 5-7. Please provide information in the legends on the significance of the green line and the faded blue lines in each partial dependence plot so the reader can understand what you are presenting here.

Line 333. Misspelling of “Keyel”.

Lines 342-345. This paragraph is hard to follow. How does the similar performance of the full and reduced models (first part of the paragraph) relate to variation in covariate data (second part)?

7. PLOS authors have the option to publish the peer review history of their article (what does this mean?). If published, this will include your full peer review and any attached files.

Reviewer #1: No

Reviewer #2: No

---

## [Editor Report · Acceptance letter]

28 Dec 2023

PONE-D-23-17817R1 

PLOS ONE

Dear Dr. Ge, 

I'm pleased to inform you that your manuscript has been deemed suitable for publication in PLOS ONE. Congratulations! Your manuscript is now being handed over to our production team.

Kind regards, 

on behalf of

Dr. José Ramos-Castañeda 

Academic Editor

PLOS ONE